

# Tibial cortex transverse transport potentiates diabetic wound healing *via* activation of SDF-1/CXCR4 signaling

Shuanji Ou[1,*], Xiaodong Wu[1,*], Yang Yang[1], Changliang Xia[1], Wei Zhang[1], Yudun Qu[1], Jiaxuan Li[1], Bo Chen[2], Lilin Zhu[1], Changpeng Xu[1] and Yong Qi[1]

[1] Department of Orthopaedics, The Affiliated Guangdong Second Provincial General Hospital of Jinan University, Guangzhou, China
[2] Department of Endocrinology, The Affiliated Guangdong Second Provincial General Hospital of Jinan University, Guangzhou, China
[*] These authors contributed equally to this work.

## ABSTRACT

**Background**. The current treatments for diabetic foot ulcers have disadvantages of slow action and numerous complications. Tibial cortex transverse transport (TTT) surgery is an extension of the Ilizarov technique used to treat diabetic foot ulcers, and can shorten the repair time of diabetic foot ulcers. This study assessed the TTT technique for its effectiveness in healing diabetic foot ulcer skin lesions and its related molecular mechanisms.

**Methods**. Diabetic rat models were established by injecting healthy Sprague-Dawley rats with streptozotocin (STZ). The effects of TTT surgery on the model rats were assessed by recording changes in body weight, analyzing skin wound pictures, and performing H&E staining to assess the recovery of wounded skin. The numbers of endothelial progenitor cells (EPCs) in peripheral blood were analyzed by flow cytometry, and levels of CXCR4 and SDF-1 expression were qualitatively analyzed by immunofluorescence, immunohistochemistry, qRT-PCR, and western blotting.

**Results**. Both the histological results and foot wound pictures indicated that TTT promoted diabetic wound healing. Flow cytometry results showed that TTT increased the numbers of EPCs in peripheral blood as determined by CD34 and CD133 expression. In addition, activation of the SDF-1/CXCR4 signaling pathway and an accumulation of EPCs were observed in skin ulcers sites after TTT surgery. Finally, the levels of SDF-1 and CXCR4 mRNA and protein expression in the TTT group were higher than those in a blank or fixator group.

**Conclusion**. TTT promoted skin wound healing in diabetic foot ulcers possibly by activating the SDF-1/CXCR4 signaling pathway.

## INTRODUCTION

A diabetic foot ulcer is one of the most serious complications of diabetes that causes disability or death, and a persistent skin ulcer is frequently found in diabetic patients (*Cho et al., 2018*). The skin wound-healing process of diabetic foot ulcers is both complex

Corresponding authors
Changpeng Xu, gd2hxcp@163.com
Yong Qi, yongqi4040@163.com

and dynamic. It is characterized by an accumulation of senescent cells and a protracted inflammatory phase accompanied by pro-catabolic balance arrest during the wound's proliferative phase, and delayed re-epithelialization (*Rodriguez-Rodriguez et al., 2022*). Even if the ulcer heals, it often relapses. Peripheral artery disease and vascular factors have been reported to increase the risk for a nonhealing ulcer (*Armstrong, Boulton & Bus, 2017*).

The current methods used to treat diabetic foot ulcers include wound debridement, pressure unloading, revascularization, and infection management (*Perez-Favila et al., 2019*). TTT surgery is an extension of the Ilizarov technique. Ilizarov first proposed the "tension-stress rule" in the mid-20th century and developed the "Ilizarov tibial transverse bone transport" technique (TTT), which promoted tissue regeneration by providing a certain amount of stress, while the patient's bones and their attached muscles, fascia, blood vessels, and nerves grew in synchrony (*Qu, 2020*). *Qu, Wang & Tang (2001)* used the TTT technique for the first time in China to treat patients with thromboangiitis obliterans of their lower limbs. The resultant case report described numerous blood vessels surrounding the bone block, which had been formed by the TTT operation. The TTT technique has also been widely used in combination with other techniques such as nose to ring draining (*Yu et al., 2021*), closed-end negative pressure drainage therapy (*Liu et al., 2022*), and antibiotic bone cementing (*Wang et al., 2020*) to treat chronic ischemic limb diseases such as vascular occlusion, diabetes, foot and ankle infections, and lower extremity ischemic diseases. TTT has been shown to accelerate wound healing by enhancing angiogenesis and immunomodulation in a diabetic foot (*Ou et al., 2022*). Compared with other therapeutic methods, TTT has the following advantages: simplicity, easy to perform, minimal surgical wound creation, and rapid subsequent wound improvement. The TTT technique can improve ischemia, relieve pain, promote wound repair, and significantly improve the percentage of salvageable limbs (*Liu et al., 2022*).

Although TTT has been successfully used in the treatment of diabetic foot ulcers, until now, only limited data has been available regarding the molecular mechanism by which TTT improves diabetic foot ulcers. Animal studies have shown that TTT surgery promotes tibial microvascular regeneration in normal domestic dogs (*Cao et al., 2019*). *Yang et al. (2022)* found that the TTT technique could accelerate wound skin healing in normal SD rats by promoting the generation of new blood vessels and causing anti-inflammatory cells to accumulate in the wounded skin. However, those study results have not been verified in a diabetic animal model.

SDF-1 (stromal cell-derived factor-1 [SDF-1]) is a homeostatic CXC chemokine, that along with its receptor (CXC chemokine receptor 4 [CXCR4]), forms a distinct signaling pathway. Studies have shown that the SDF-1/CXCR4 signaling pathway plays a critical role in wound healing. In addition, the SDF-1/CXCR4 pathway can activate major physiological processes associated with wound healing, such as an inflammatory response to tissue damage, cell proliferation, collagen deposition needed for tissue remodeling, and an increase in angiogenesis in targeted diseases (*Chen et al., 2021a*). Therefore, we speculated that TTT surgery might promote the wound healing of diabetic foot ulcers and activate the SDF-1/CXCR4 signaling pathway, simultaneously. Our study explored the

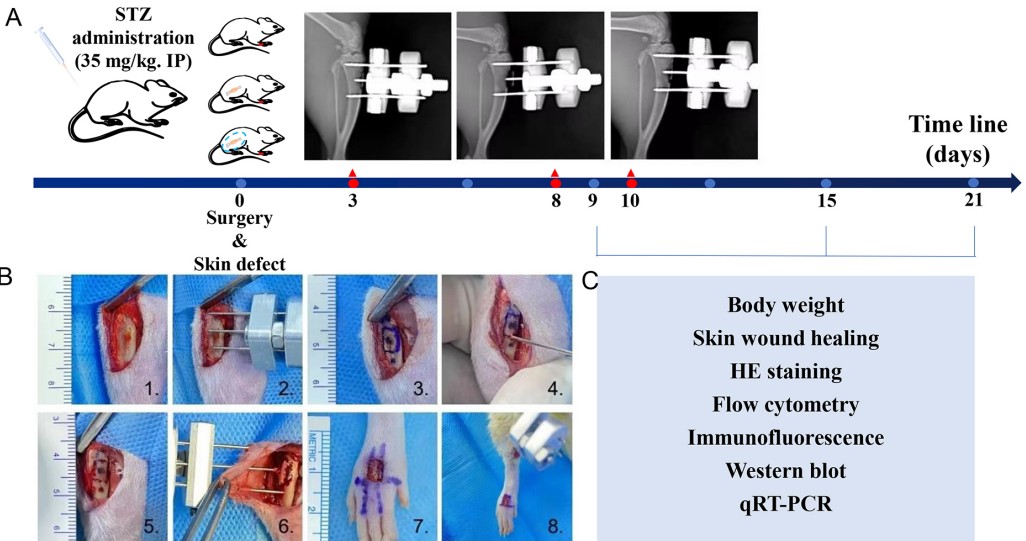

**Figure 1** **Introduction of diabetic rats model, tibial transverse bone transport (TTT) surgical procedure and workflow.** (A) Diabetic rats model. (B) Surgical procedure of TTT technique and skin defect: (1) incision of skin and tibial was exposed; (2) pre-assembled external fixator frame; (3–5) tibial bone window; (6) the cortical bone chip was dislocated, and the external fixator frame was attached and skin sutured; (7) and (8) the full-thickness skin wound of four mm * five mm was performed. (C) Experimental tests on specific dates.

biological mechanism by which the TTT technique promotes the healing of diabetic foot ulcers.

## MATERIALS AND METHODS

### Diabetic rats

A total of 54 male SD rats (age, 6-weeks; weight range, 280–410 g) were used to establish the diabetic rat model. Diabetes was induced by intraperitoneal injection of 35 mg/kg STZ (YuanYe Bio, Shanghai, China) as described in a previous report (*Furman, 2021*). The rats were allowed acclimate to their new environment for at least 5 weeks prior to being injected with STZ. The rats were housed (five rats per cage) in a room maintained at $21 \pm 1\,°C$, $55 \pm 5\%$ humidity, and with a 12-h light dark cycle (08.00 on: 20.00 off). The rats had *ad libitum* access to water and a high fat diet (formula details: 10% lard, 2.5% cholesterol, 20% sucrose, 1% cholate, and 66.5% common diet, respectively), for at least one week; after which they were fasted for 12 h prior to being injected with STZ. On the last day, after being fasted for 12 h, a sample of blood was obtained from the caudal vein of each rat and used to measure the blood glucose level. A blood glucose level >16.7 mmol/L indicated that the diabetic rat model was successfully established (Fig. 1A). The rats were obtained from the Animal Center of Southern Medical University and the protocols for all animal studies were approved by an Institutional Review Committee for the use of Animal Subjects, and conformed with guidelines established by the Second People's Hospital of Guangdong Province Experimental Animal Ethics Committee (No. 2022-dw-kz-006-01).

## Surgical procedures

The diabetic rats were allowed to recover for at least one week prior to being used in any experiments. All experimental rats were anesthetized and operated in a sterile operating room to minimize postoperative complications. The rats with successful diabetes modeling were divided into three groups: a sham operation group, tibial osteotomy group, and TTT group, respectively. A total of 18 diabetic rats that were not implanted or received a sham operation served as a negative control group, designated as the blank group. The tibial osteotomy group consisted of 18 diabetic rats that underwent fixator implantaton surgery; that group was designated as the fixator group. A total of 18 diabetic rats implanted with the TTT fixator formed the TTT group. The TTT procedure was performed as described in a previous study (*Chen et al., 2021a*; *Furman, 2021*) (Fig. 1B 1–6). The TTT procedure for rats required nine days to complete, and. the middle and upper segments of the tibia were selected for cortical osteotomy. Guided with the external fixator, a pneumatic saw was used to penetrate the bone cortex to form a corticotomy window of 10 mm × five mms. After punching holes, steel needles were passed through the cortex, fixed in the holes and then assembled to form an external fixator. Two 2-mm threaded needles were used to perform cortical bone chip transport. After surgery, the animals were allowed to rest for 3 days. On day 4 after surgery, the bone chip was transversely pulled outward by 0.25 mm every 12 h for three days, and then pulled inward in the same manner during the last three days. All rats were operated on the dorsum of the foot to form a full-thickness skin wound of four mm × 5 mm (Fig. 1B 7–8), No obvious signs of infection or other complications were observed in any of the experimental animals after the operation. The surgery day was designated as day 0. On the third day, the postoperative extension was extended for 5 days, stopped for 2 days, and then pulled back for 5 days in the TTT group. All 6 diabetic rats in each group were euthanized using a $CO_2$ euthanasia device (BM Shanghai Biowill Co., LTD., Shanghai, China) with a flow of 6.5 L/min on days 9, 16, and 21 after surgery, respectively. Changes in body weight were recorded, pictures of skin lesions were taken, and pathological examinations were performed (Fig. 1C).

## Body weight

As mentioned above, all rats were subjected to uniform rearing standards. To limit variations in baseline data, body weights were recorded prior to injection of STZ, and diabetic rats were weighed a second time following successful modeling. Each rat was weighed prior to the procedure, and after the procedure, weight changes in the groups were monitored and recorded.

## Skin wound recovery

Following surgery, the experimental rats were fed under normal conditions, and the ulcer wounds were photographed at various intervals.

## Hematoxylin and eosin (H&E) staining

Morphological changes in skin wound tissue of the rat dorsal foot wounds were detected by H&E staining on days 9, 15, and 21 after surgery. The paraformaldehyde-fixed skin tissues were placed in a descending alcohol series; after which, they were dehydrated,

fixed, and paraffin-embedded. The blocks of tissue were then cut into 3 μm-thick sections with a microtome. Next, the sections were deparaffinized, rehydrated, and immersed in hematoxylin for 5-8 min before being washed with phosphate-buffered saline (PBS). The sections were then immersed in 0.1% acid water and subsequently exposed to 1% eosin for 1-3 min. Finally, the sections were treated with an ascending alcohol series, cleared with xylene, mounted onto slides, and observed and photographed under a light microscope (NIKON, ECLIPSE Ci, Japan).

## Immunofluorescence (IF)

As noted above, the prepared skin sections were deparaffinized and rehydrated prior to antigen retrieval. Following rehydration, the sections were placed in an enclosed plastic container, rinsed three times in 100 mL PBS for 5 min, and then subjected to antigen retrieval with EDTA (pH 8.0). A circle was draw around the tissues with an IHC PAP pen. The sections were blocked with blocking buffer for at least 30 mins at room temperature after antigen retrieval. The sections were then removed from the blocking buffer, treated with diluted primary antibodies, and incubated overnight in a wet chamber at 4 °C. The next morning, the tissues were incubated with a secondary antibody for 50 mins at room temperature. Finally, the nuclei were counterstained with 4,6-diamidino-2-phenylindole (DAPI; 0.1 μg/mL). The antibodies used in the study were anti-CD34 (sc-18917,1:200; Santa Cruz Biotechnology, Dallas TX, USA), anti-CD133 (sc-365537, 1:200; Santa Cruz Biotechnology), anti-CXCR4 (ab124824; Abcam, Cambridge, MA, USA, 1:200), anti-SDF-1 (17402-1-AP; Proteintech, Rosemont, IL, USA, 1:500), Goat anti-rat-IgG H&L (FITC, ab6840, 1:1,000; FITC, Abcam), Goat anti-mouse-IgG H&L (Alexa Fluor®594, ab150120, 1:200; Abcam), and Goat anti-rabbit IgG H&L (Cy5®, ab6564, 1000; Abcam).

## Immunohistochemistry (IHC)

Sections of skin tissue were blocked with 3% $H_2O_2$ for 10 min, washed three times in PBS, permeabilized with 0.2%Triton for 10 min, and then incubated overnight with primary antibodies at 4 °C, followed by incubation with secondary antibodies for 30 min at room temperature. Next, DAB chromogenic solution (P0202; Beyotime, Jiangsu, China) was added to the slices, and the reaction was terminated by immersion in PBS. Finally, the sections were stained with hematoxylin, fixed with 95% alcohol, sealed with neutral resin, and observed and photographed under a light microscope. Anti-CXCR4 (Abcam, ab124824, 1:200) was used in this IHC procedure.

## Flow cytometry

In order to detect changes in the numbers of EPCs in peripheral blood, CD34 and CD133 were used as markers of EPCs. Samples of orbital blood were collected into anticoagulant tubes and diluted with PBS. Next, reagent A and reagent D were added to form a gradient as described in instructions for use of peripheral blood monocyte cell separation liquid (P6700; Solarbio). Gradient centrifugation was performed to filter the mixture at room temperature, and the white monocyte layer was transferred into a 15 mL centrifuge tube and washed with PBS. Next, primary antibodies were added to the monocyte cells and incubated at 4 °C for 60 min at room temperature. After washing with PBS, the cells were

Ou et al. (2023), *PeerJ*, DOI 10.7717/peerj.15894

**Table 1  Primers used for qRT-PCR.**

| ID | Primer | Sequence (5′–3′) |
| --- | --- | --- |
| GAPDH | F | TTCAACGGCACAGTCAAG |
| GAPDH | R | TACTCAGCACCAGCATCA |
| CXCR4 | F | CTGTGGATGGTGGTGTTC |
| CXCR4 | R | AGGAAGGCGTAGAGGATG |
| SDF1 | F | GCATCAGTGACGGTAAGC |
| SDF1 | R | GCATCAGTGACGGTAAGC |

**Notes.**

F, Forward Primer; R, Reverse Primer.

incubated with secondary antibodies in the dark at 4 °C for 30 min. Finally, pre-cooled PBS was added to the monocytes and the cells were analyzed by flow cytometry (CytoFlex; Beckman Coulter). The antibodies used for flow cytometry studies were anti-CD133 (18470-1-AP, 0.2 µg/mL; Proteintech), anti-CD34 (60180-1-Ig, 0.2 µg/mL; Proteintech), goat anti-Rabbit IgG H&L (PE) (ab72465, 1:500; Abcam), and goat anti-mouse IgG H&L (FITC) (ab6785, 1:500; Abcam).

## Real-time fluorescence quantitative PCR (qRT-PCR)

qRT-PCR was performed to detect the expression of SDF-1 and CXCR4 in the skin wound tissues of diabetic rats. Total RNA was extracted from skin wound tissue using TRIzol reagent (Invitrogen, Carlsbad, CA, USA), and cDNA was synthesized using a PrimeScript™ RT reagent Kit (Takara, Shiga, Japan) as described in the manufacturer's instructions. qRT-PCR was performed by using SYBRR Green Realtime Master Mix (Takara) on an ABI Real time PCR (7500) instrument. (with) The cycle conditions used for qRT-PCR were as follows: initial incubation for 3 mins at 95 °C, followed by 35 cycles of 5 s at 95 °C and 5 s at 60 °C. Relative levels of gene expression were calculated using the $2^{-\Delta\Delta CT}$ method. The primers used for qRT-PCR are listed in Table 1.

## Western blotting (WB)

A protein extraction kit (Beyotime, P0013J) was used to extract the total protein from rat skin ulcer tissues according to the manufacturer's protocol. Prior to centrifugation at 12,000g for 15 min at 4 °C, TissueLyser beads were added to the centrifugation tubes to assist with protein cleavage. The upper layer of lipid was removed, and a BCA Protein Assay Kit (Beyotime, China) was using to determine the protein concentration in each extract. Western blotting was performed according to a standard WB protocol (*Yukhananov, Chimento & Marlow, 2022*). ImageJ software (v 1.8.0) was used to quantify protein levels, and GAPDH expression served as an internal standard. The primary antibodies used for WB were as follows: anti-CXCR4 (ab124824, 1:100; Abcam), anti-SDF1 (ab25117, 1:1,000; Abcam), and anti-GAPDH (ab9485, 1:2500; Abcam). The secondary antibody was goat anti-rabbit IgG H&L (HRP) (ab6721, 1:2000; Abcam).

## Statistical analysis

All data were analyzed using IBM SPSS Statistics for Windows, Version 25 software (IBM Corp., Armonk, NY, USA) and graphs were drawn using GraphPad Prism 8.0 software.

One-way ANOVA followed by Tukey's post hoc test or the student's t test was used to compare mean values among different groups. Quantitative data are expressed as a mean value $\pm$ SEM. A $p$-value $< 0.05$ was considered to be statistically significant.

## RESULTS

### TTT accelerated wound closure and promoted skin tissue recovery

Measurements of rat body weight taken before and after STZ injection revealed no significant change in body weight. However, there were significant differences in body weight among the three different groups (blank group *vs.* Fixator group *vs.* TTT group) at 9 days after the operation (Fig. 2B). Especially at 21 days following surgery, there was no discernible difference in body weight among the different treatment groups. In order to observe the healing of the ulcer after operation, the wounds were photographed on days 3, 6, 9, 12, and 15 after surgery. H&E staining was performed on days 8, 15, and 21 after surgery. When compared to the rats in the blank group, the wound healing areas of rats in the TTT group were the smallest, followed by the wound healing areas in the Fixator group, especially at 15 days after the operation (Fig. 2C). H&E staining was used to examine the regenerative capability of wounded skin (Fig. 2D). In the blank group, the cells of the wounded skin were discontinuous and broken, even at 21 days after the operation. After treatment with the TTT technique, the epidermal and dermal cells of wound skin were more closely and orderly arranged, and the thickness of the dermis was greater than that in the Fixator group. When taken together, these findings indicated that the TTT technique not only significantly accelerated the wound healing process, but also enhanced skin tissue regeneration and repair.

### TTT increased the levels of EPCs to contribute to wind healing

The levels of EPCs in peripheral blood were detected by flow cytometry and using CD34 and CD133 as markers (Fig. 3A). Our data showed that at different time points after operation, the TTT technique significantly increased EPC levels when compared to levels in the blank group, and the differences became increasingly obvious over time. The numbers of CD34+CD133+ cells in the Fixator group were significantly greater than those in the blank group only at 21 days after surgery (Fig. 3B, $p < 0.001$). As the post-surgery time increased, the difference in EPC numbers between the TTT group and Fixator group steadily widened, indicating that TTT had a more rapid and pronounced effect.

### TTT increased the numbers of EPCs and activated the SDF-1/CXCR4 signaling pathway

EPC levels were detected by the immunofluorescence of CD34 and CD133, and the deposition of SDF-1 and CXCR4 proteins was detected using immunofluorescence and immunohistochemistry; those results are summarized in Figs. 4–6. Our data showed that three indicators (CD34, CD133. and CXCR4) had increased expression levels in the TTT group, regardless of the time after surgery (Fig. 4A) and an IF quantitative analysis showed the same result (Figs. 4B–4D). The levels of fluorescence intensity in the fixator group were significantly higher than those in the blank group on days 15 and 21 after surgery.

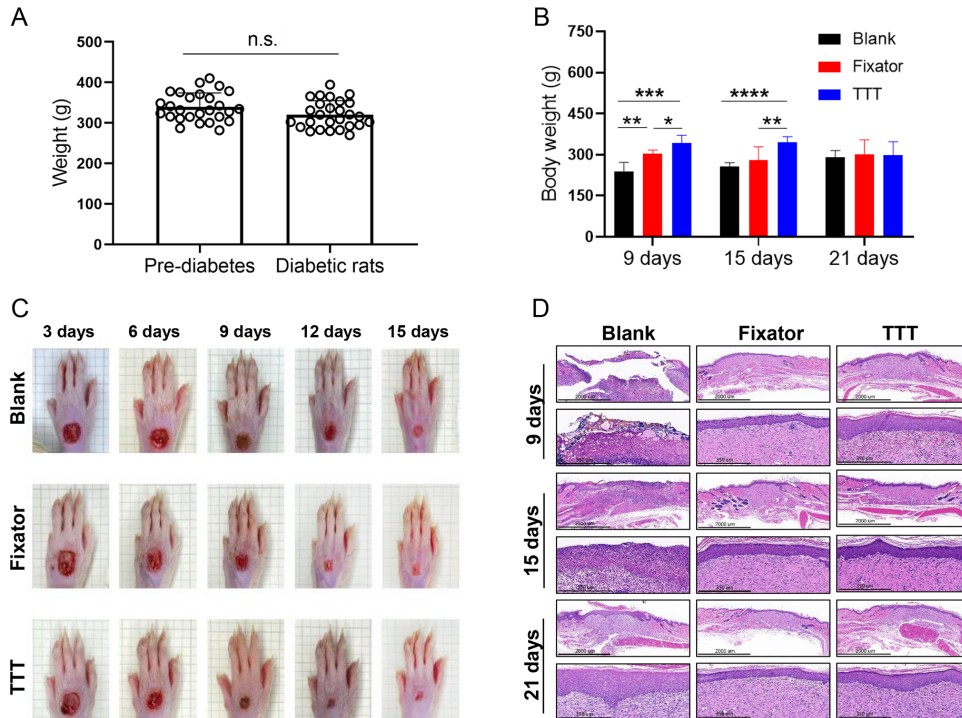

**Figure 2 The effects of tibial transverse transport (TTT) technique on body weight and skin recovery in diabetic rats.** (A) Weight change of rats undergoing diabetes modeling ($N = 27$, n.s. $P > 0.05$). (B) The changes of body weight on diabetic rats after different operations ($N = 6$, *$P < 0.05$, **$P < 0.01$, ***$P < 0.001$, ****$P < 0.0001$). (C) Representative images of wound healing progress for three groups from the same SD rats. (D) Morphological images of HE staining for ulcer skin tissues treated with different operations.

Similar trends in CD34, CD133, and SDF-1 expression are shown in Fig. 5A. Moreover, a quantitative examination of IF results was also conducted (Figs. 5B–5D). However, those results showed relatively higher fluorescence intensities occurring in the TTT group at 15 days after surgery rather than at 21 days after surgery.

Our results showed that CXCR4 was highly expressed in skin wounds in the TTT group and the levels of CXCR4-positive cells increased over time. On day 21 after surgery, high numbers of CXCR4 positive cells were found in both the fixator group and TTT group (Fig. 6A).

TTT treatment significantly increased the numbers of CXCR4-positive cells in the skin wounds of diabetic rats. Meanwhile, our statistical results were consistent with the phenomenon observed by IF (Fig. 6B).

## TTT promoted SDF-1/CXCR4 expression

qRT-PCR and western blotting were used to measure SDF-1/CXCR4 expression in skin tissue, both quantitatively and qualitatively. Our data showed that TTT treatment increased the levels of CXCR4 expression in skin wound sites (Fig. 7A). Significant differences in SDF-1 mRNA expression between the TTT group and blank group were not found until
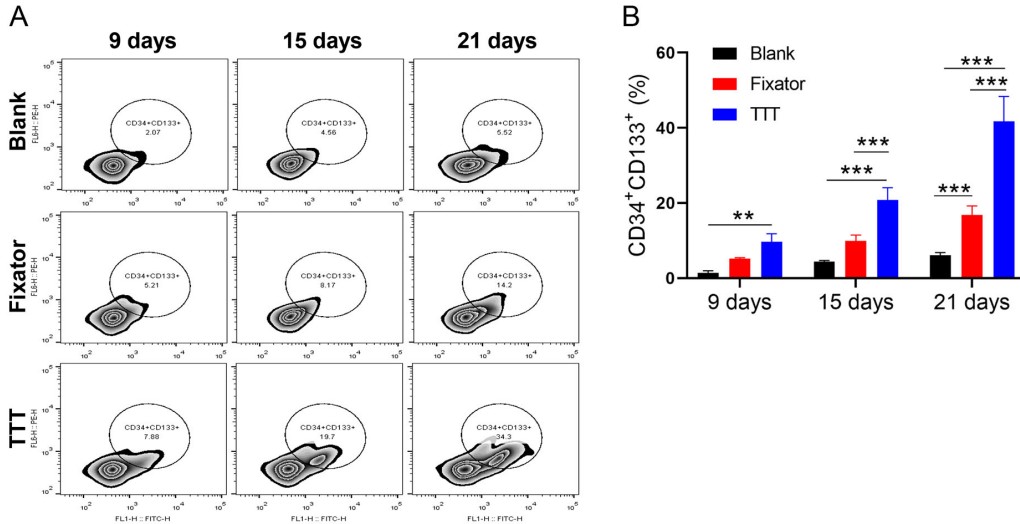

**Figure 3** **TTT promoted the level of EPCs.** (A) The level of endothelial progenitor cells was evaluated with using flow cytometry. (B) Monitoring CD34/CD133 levels every six days in wound skin *via* using flow cytometry, the results showed the highest levels of hematopoietic stem cell in TTT group, especially at 21st days after operation (**$P < 0.01$, ***$P < 0.001$).

21 days after surgery (Fig. 7B). The effects of TTT surgery on SDF-1/CXCR4 protein expression in skin wound sites were consistent with those shown by mRNA analysis (Fig. 7C). High levels of SDF-1/CXCR4 proteins were expressed in the TTT group on days 9, 15, and 21 after surgery when compared to their expression in the blank group. On days 9 and 21 after surgery, the levels of CXCR4 and SDF-1 proteins were significantly increased in the fixator group when compared with those in the blank group (Figs. 7C–7E). Both the quantitative and qualitative experimental results showed that the levels of SDF-1/CXCR4 protein expression in the TTT group were significantly higher than those in the fixator or blank group. This indicated that the TTT technique contributed to skin wound healing by activating the SDF-1/CXCR4 signaling pathway. Overall, these findings showed that TTT promoted skin wound healing over time.

## DISCUSSION

This study is one of the few to date that used a diabetic animal model to confirm the impact of TTT surgery on diabetic foot ulcers, and also explore the relevant molecular mechanism. The TTT technique contributes to skin wound healing by accelerating the regeneration of ulcer skin, while simultaneously activating the SDF-1/CXCR4 signaling pathway.

Diabetic foot ulcers are a medical challenge for the growing community of people with diabetes throughout the world. *Burgess et al. (2021)* pointed out that several procedures (debridement, oxygen therapy, off-loading, and skin substitutes) could be used to treat diabetic foot ulcers. Moreover, he stated that postoperative hyperglycemic conditions could influence the normal immune response, making the wound healing process susceptible to infection or aggravation of an existing infection, and thereby delay wound healing. Other

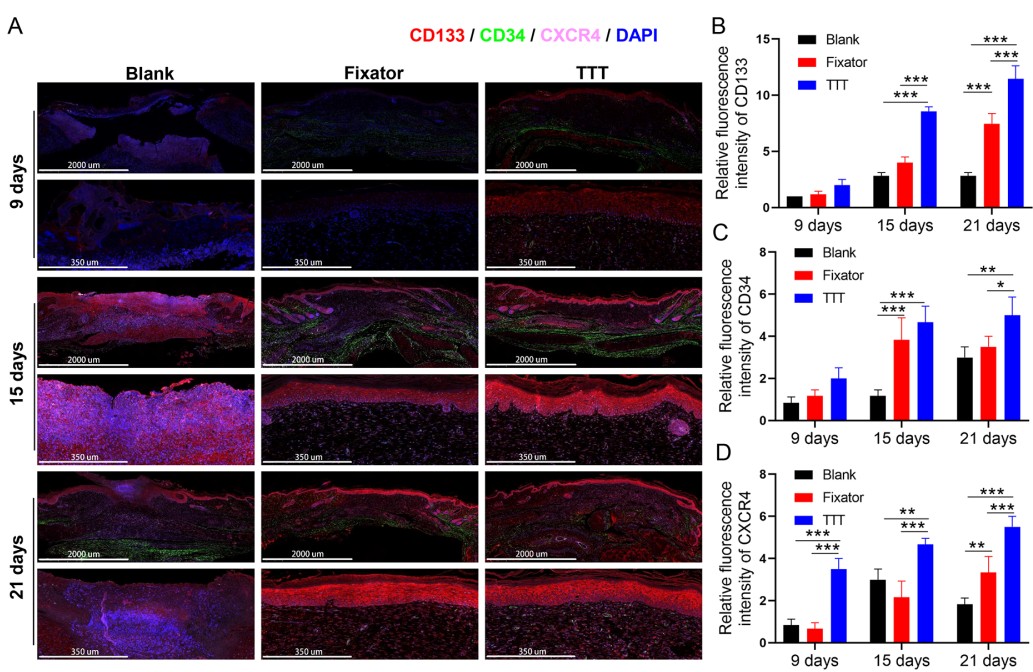

**Figure 4** **TTT could effectively increase CD34/CD133 and CXCR4 expression in wound skin.** (A) Immunofluorescence analysis of CD34/CD133 indicated that TTT stimulated neovascularization in wound skin (CD34 was stained green, CD133 was stained red). Chemokine was marked by CXCR4 (pink) in wound skin from each group. Nuclei were stained with 4′, 6-Diamidino-2-Phenylindole (DAPI, blue). (B–D) Relative fluorescence intensity of CD133, CD34 and CXCR4 were quantitatively analyzed (*$P < 0.05$, **$P < 0.01$, ***$P < 0.001$).

treatments for diabetic foot ulcers include negative-pressure wound therapy, energy-based therapy, stem cell therapy, dressings, and topical agents (*Everett & Mathioudakis, 2018*). Negative-pressure wound therapy has been found to be effective for promoting wound closure, but its use is limited by its high cost (*Chen et al., 2021b*). Other treatments may be associated with a risk for amputation and wound recurrence, and there is inadequate evidence to recommend any those therapies (*Everett & Mathioudakis, 2018*). Therefore, investigators need to explore new techniques for reducing the rates of limb amputation and ulcer recurrence. It is well known that certain parts of the foot with reduced perception experience continuous mechanical and shear stress (*De Wert et al., 2019*; *Jones et al., 2022*). The TTT technique applies a slow, steady, and long-lasting stimulus to bone tissue, muscles, and nerves *via* mechanical stretching to promote the regeneration of blood vessels and bone marrow and improve blood circulation; these effects ultimately restore limb perception (*Zuo et al., 2018*). Other studies have demonstrated that mechanically stretched skin exhibits the migration of bone marrow mesenchymal cells, and normal donor skin can be mechanically stretched to promote its regeneration (*Zhou et al., 2013*). An Expert Consensus on the treatment of diabetic foot ulcers using TTT reported that TTT could stimulate microcirculation and nerve function recovery in the lower limbs of a diabetic foot and significantly reduce the wound size and overall risk for diabetic foot complications when compared with other treatment methods (*Chinese Bone Transport Diabetic Foot Group,*

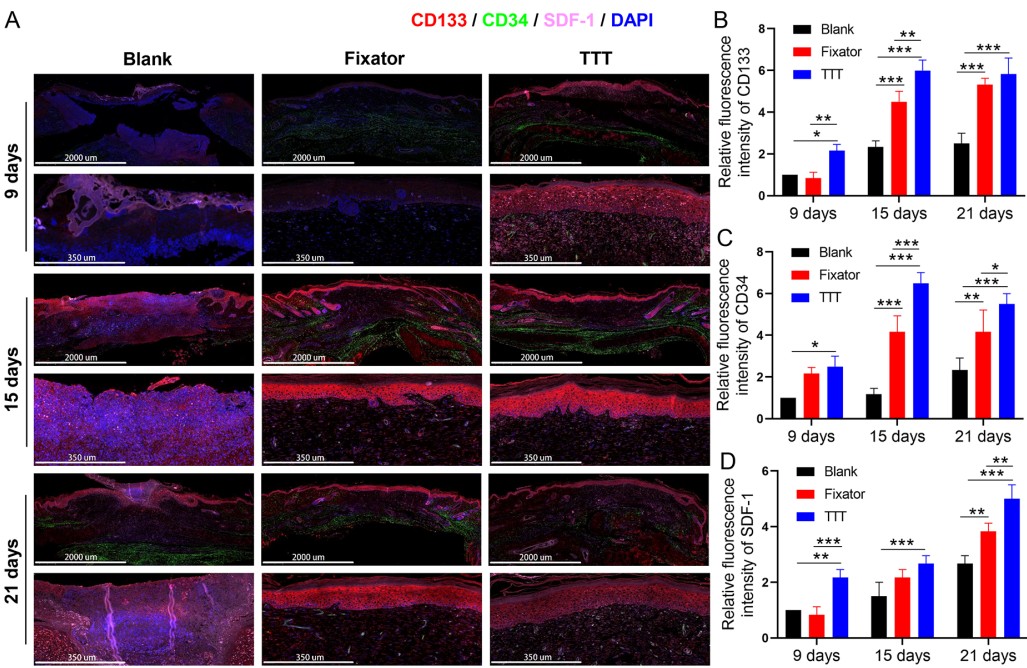

**Figure 5** **TTT could effectively increase CD34/CD133 and SDF-1 expression in wound skin.** (A) Immunofluorescence analysis of CD34/CD133 indicated that TTT stimulated neovascularization in wound skin (CD34 was stained green, CD133 was stained red). Chemokine was marked by SDF-1 (pink) in wound skin from each group. Nuclei were stained with 4′, 6-Diamidino-2-Phenylindole (DAPI, blue). (B-D) Relative fluorescence intensity of CD133, CD34 and SDF-1 were quantitatively analyzed ($*P < 0.05$, $**P < 0.01$, $***P < 0.001$).

*2020*). Methods for treating diabetic foot ulcers need to be simultaneously evaluated for their effectiveness, safety, and cost effectiveness. Claims that TTT might hasten the recovery of severe and recalcitrant diabetic foot ulcers are not true, and there is no denying that TTT can produce adverse effects such as secondary fractures, skin necrosis at the surgical site, pin tract infection during transport, edema, pain, and bleeding (*Zhang et al., 2020*).

One previous study explored the biochemical mechanism of TTT in diabetic mice. The use of STZ to create a diabetic rat model has been demonstrated in multiple studies (*Furman, 2021*; *Pandey & Dvorakova, 2020*). Furthermore, our study employed the TTT operation, which is a modified Ilizarov technique (*Zhao et al., 2020*), to quickly accumulate a large number of EPCs (CD33/CD134 positive cells) at just 9 days after the operation, which significantly shortened the wound healing time. Our IF and IHC experiments did not reveal a significant accumulation of SDF-1/CXCR 4 in the skin ulcer tissues of the rats. In addition to releasing cytokines that promote the vascularization of diabetic skin wounds, it has been demonstrated that EPCs can increase the levels of SDF-1/CXCR4 mRNA and protein expression. That effect is closely related to the biological functions of EPCs

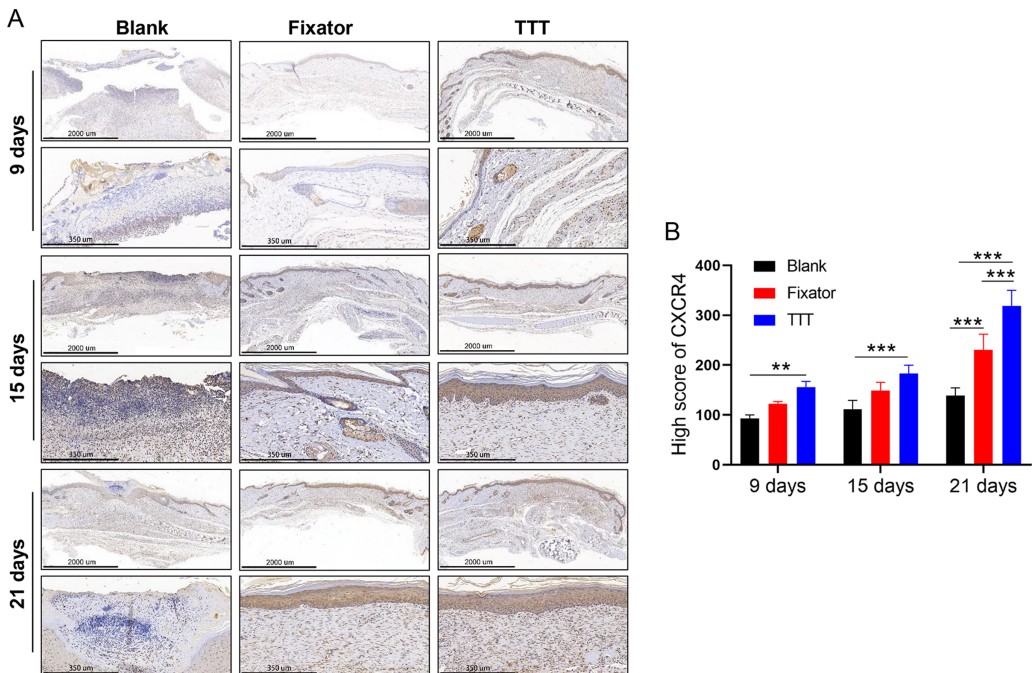

**Figure 6** **TTT activated CXCR4 signal pathway and promoted cell apoptosis.** (A) Immunohistochemistry analysis of CXCR4 positive cells (brown) in skin wound from each group. (B) High score of CXCR4 were quantitatively analyzed (**$P < 0.01$, ***$P < 0.001$).

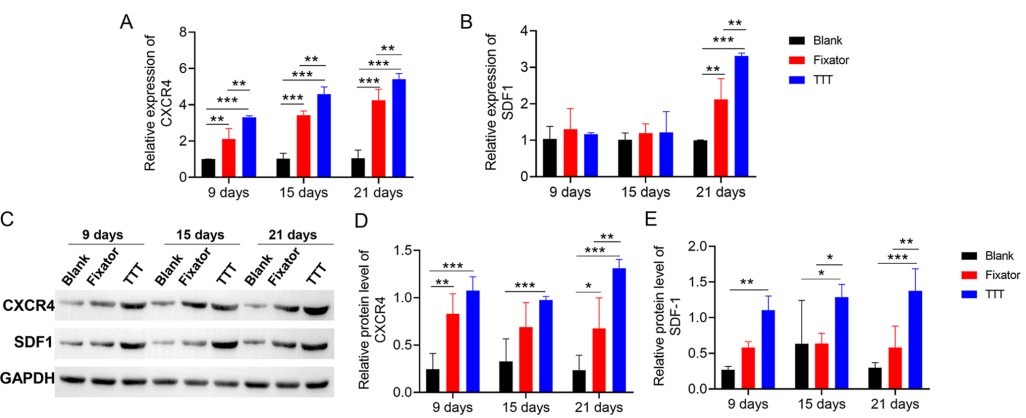

**Figure 7** **TTT promoted mRNA and protein expression of CXCR4 and SDF-1.** (A) The mRNA expression of CXCR4 in the wound skin (**$P < 0.01$, ***$P < 0.001$). (B) The mRNA expression of SDF-1 in the wound skin (**$P < 0.01$, ***$P < 0.001$). (C) Western blotting analysis of the CXCR4 and SDF-1 in skin tissues of each group. GAPDH was used as an internal parameter. (D) Corresponding quantification of band density of CXCR4. (E) Corresponding quantification of band density of SDF-1 (*$P < 0.05$, **$P < 0.01$, ***$P < 0.001$).

(*Leng et al., 2021*), and further validates the results in our study. In addition, we also found that the degree of skin healing in the TTT group at 21 days was significantly greater than that in the blank group, which is consistent with previous studies (*Lian, 2019*). It is well known

that, treatment of diabetic foot ulcers using the TTT technique is based on mechanical stretching. *Zhou et al. (2013)* and *Anh (2017)* reported that mechanical stretching could upregulate SDF-1 expression and activate the SDF-1/CXCR4 signaling pathway to cause the migration of mesenchymal stem cells to areas of skin regeneration. Another study showed that SDF-1/CXCR4 interactions were critical for skin re-epithelialization, as they could accelerate skin epithelialization and dermal structural regeneration by promoting fibroblast migration during the remodeling stage of skin healing (*Yari et al., 2020*). Our results suggest that TTT promotes hemostasis, EPC migration, and skin regeneration by activating the SDF-1/CXCR4 signaling pathway, to accelerate the healing of diabetic foot ulcers. Wound healing is a process that consists of four stages: hemostasis, inflammation reduction, vascularization, and remodeling. Dysfunction during any of these stages will affect the healing of diabetic wounds (*Patel et al., 2019*). CXCR4 had been proven to participate in immune regulation and control the regeneration of multiple organs and tissues (*Bianchi & Mezzapelle, 2020*), suggesting that the anti-inflammatory effect of the TTT technique during the healing process of diabetic foot ulcers results from activation of the SDF-1/CXCR4 signaling pathway.

Although our study demonstrated that the TTT technique plays an important role in improving microcirculation reconstruction, reducing inflammation, and accelerating wound healing, we should not ignore the limitations of the study. First, we did not extend the observation time of skin healing beyond 21 days after surgery. Second, the mechanism by which TTT promotes skin healing was not fully elucidated; for example, the mechanism involved in microcirculation reconstruction and inflammation reduction in diabetic foot ulcers was not verified. Finally, in order to confirm the precise function of the SDF-1/CXCR4 signaling axis, it will be necessary to perform reverse verification by use of by agonists or inhibitors.

### Funding

This work was supported by the National Natural Science Foundation of China (No. 81972083), Science and Technology Planning Project of Guangzhou (Nos. 202102080052, 202102010057 and 202201020303), and the Science Foundation of Guangdong Second Provincial General Hospital (Nos. YQ2019-009, 3D-A2020002, and 3D-A2020004). The funders had no role in study design, data collection and analysis, decision to publish, or preparation of the manuscript.

### Grant Disclosures

The following grant information was disclosed by the authors:
National Natural Science Foundation of China: 81972083.
Science and Technology Planning Project of Guangzhou: 202102080052, 202102010057, 202201020303.
Science Foundation of Guangdong Second Provincial General Hospital: YQ2019-009, 3D-A2020002, 3D-A2020004.

## Competing Interests

The authors declare there are no competing interests.

## Author Contributions

- Shuanji Ou conceived and designed the experiments, performed the experiments, prepared figures and/or tables, authored or reviewed drafts of the article, and approved the final draft.
- Xiaodong Wu conceived and designed the experiments, authored or reviewed drafts of the article, and approved the final draft.
- Yang Yang performed the experiments, authored or reviewed drafts of the article, and approved the final draft.
- Changliang Xia performed the experiments, authored or reviewed drafts of the article, and approved the final draft.
- Wei Zhang analyzed the data, authored or reviewed drafts of the article, and approved the final draft.
- Yudun Qu analyzed the data, authored or reviewed drafts of the article, and approved the final draft.
- Jiaxuan Li analyzed the data, prepared figures and/or tables, and approved the final draft.
- Bo Chen performed the experiments, prepared figures and/or tables, and approved the final draft.
- Lilin Zhu performed the experiments, analyzed the data, prepared figures and/or tables, and approved the final draft.
- Changpeng Xu conceived and designed the experiments, prepared figures and/or tables, and approved the final draft.
- Yong Qi conceived and designed the experiments, prepared figures and/or tables, authored or reviewed drafts of the article, and approved the final draft.

## Animal Ethics

The following information was supplied relating to ethical approvals (*i.e.*, approving body and any reference numbers):

The Second People's Hospital of Guangdong Province Experimental Animal Ethics Committee approved the study (2022-dw-kz-006-01).

## Data Availability

Data are available at Figshare:

Qi, Yong (2023). Original+Data.zip. figshare. Journal contribution. https://doi.org/10.6084/m9.figshare.22672939.v1.

## Supplemental Information

Supplemental information for this article can be found online at http://dx.doi.org/10.7717/peerj.15894#supplemental-information.

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
