# Peer review of "Tibial cortex transverse transport potentiates diabetic wound healing via activation of SDF-1/CXCR4 signaling"

_PeerJ, doi:10.7717/peerj.15894_

## Round 0.1 · original submission · Minor Revisions

This manuscript illustrates the underlying mechanism of diabetic wound healing and has a promising clinical application. However, the authors' immunohistochemistry lacks quantitative analysis, and the picture should also be supplemented with scale. In addition, the literature is slightly old and the limitations of the study are not mentioned in the discussion section.

Reviewer 1 ·

Basic reporting

1. I think the authors described an important work. This article provides a relatively clear explanation of the author's findings and intentions. If the English expression is more authentic, it will be more helpful for readers to understand.
2. Some references are too old to reflect the current information on medical research progress.
Ilizarov GA. The tension-stress effect on the genesis and growth of tissues. Part I. The influence of stability of fixation and soft-tissue preservation. Clin Orthop Relat Res. 1989 Jan;(238):249-81. PMID: 2910611
Qu L, Wang A, Tang F. The therapy of transverse tibial bone transport and vessel regeneration operation on thromboangitis obliterans. Zhonghua Yi Xue Za Zhi. 2001 May 25;81(10):622-4. Chinese. PMID: 11798937

Experimental design

3. It seems that TTT has multiple beneficial effects on diabetic wound healing, including reducing wound healing time, enhancing new skin quality, promoting the deposition of endothelial progenitor cells, and activating the SDF-1/CXCR4 signaling pathway.
4. In the Abstract, the author should describe the methods in detail stead of “The effect of TTT on the healing of diabetic foot ulcers was assessed in terms of histomorphology, molecular organism, foot wound and body weight of diabetic rats.
5. In the Introduction, the authors should introduce the advantages of Tibial cortex transverse transport (TTT) surgery to other treatments for diabetic foot ulcers to ensure the study’s value.

6. The primer sequences of SDF-1 and CXCR4 genes should be provided.

Validity of the findings

7. In general, there is a lack of quantitative analysis for the results of immunofluorescence and immunohistochemistry. Furthermore, the scale bar for each picture should be added.
8. Some agonists and antagonists of the SDF/CXCR4 signaling pathway needed to be worked in reverse.
9. The conclusions are overstated. For example, the study did not show any results on microcirculation reconstruction.

Additional comments

10. It is recommended to clarify the limitations of this study, such as the lack of clinical data support.

Reviewer 2 ·

Basic reporting

1.1 The manuscript Title:Tibial transverse bone transport potentiates diabetic wound healing by activating SDF-1/CXCR4 signal . These findings further support the potential of TTT as a treatment for diabetic foot ulcers, the authors should be commended for their efforts, as the topic is important, the approach is logical, and the results are intriguing. However, the manuscript needs careful editing by someone with expertise in technical English editing.
1.2 For researchers who are not clinical doctors, the "Ilizarov tibial transverse bone transplantation" technique (TTT) is too specialized and requires a brief description of the surgical method.
1.3 The citation of references is relatively reasonable.

Experimental design

2.1 It is better to change the title “Tibial transverse bone transport potentiates diabetic wound healing by activating SDF-1/CXCR4 signal”to “Tibial cortex transverse transport potentiates diabetic wound healing by activating SDF-1/CXCR4 signal”.

2.2 In the introduction, the author indicated that “Our study attempted to reveal the biological mechanism of TTT technique in improving diabetic foot ulcers healing.” Please insured that the study aims to clarify the biological or molecular mechanism of TTT technique in improving diabetic foot ulcers healing.

2.3 In the Introduction, In the first paragraph of the introduction, the authors indicated that “The SDF-1/CXCR4 pathway could activate the major physiological processes associated with wound healing such as inflammatory response to damage tissues, cell proliferation, collagen deposition for tissue remodelling and enhance angiogenesis in targeted diseases”. In the second paragraph of the introduction, the authors indicated that “TTT technique could accelerate wound skin healing on normal SD rats by generating new blood vessels and accumulating the level of anti-inflammatory cells in the wound skin.” It seems that the authors indicated that could not be speculated that TTT could promote the wound healing of diabetic foot ulcers by activating the SDF-1 / CXCR 4 signaling pathway.”The authors need to supplement the evidence in the introduction.
2.4 In Materials and Methods, the agents or the methods should be more detailed, such as secondary antibodies in IF, reagent A and reagent B in Flow cytometry and the detailed procedures in qRT-PCR.

Validity of the findings

3.1 Many descriptions in the Results are not precise enough, and the author needs to provide a more accurate description of these results, such as “TTT treatment significantly increased CXCR4 positive cells in the skin wound of diabetic rats, which means TTT could enhance the migration of CXCR4 induced by SDF-1.” There is no direct evidence to prove that TTT could enhance the migration of CXCR4 induced by SDF-1.
3.2 For immunofluorescence /immunohistochemistry experiments(Fig 4, Figure 5 and Figure 6), it is best to perform quantitative analysis.
3.3 In figure 7A, this result is a quantitative result of WB, so the results of the WB experiment need to be presented first.

Additional comments

4.1 In the discussion, the author claimed that “Although the research shown that TTT technique played an important role in improving microcirculation reconstruction, anti-inflammatory, and accelerating wound healing, we could not ignore limitations of the study.” The authors should indicate the limitations of the study and discuss the possible solutions.

---

## Round 0.2 · accepted · Accept

The manuscript meets the reviewer's requirements and can be accepted in its current state.

Reviewer 1 ·

Basic reporting

no comment

Experimental design

no comment

Validity of the findings

no comment

Additional comments

no comment

Reviewer 2 ·

Basic reporting

The author's verification and revision of the suggestions were positive and effective, and the overall quality of the manuscript was improved enough to endorse its publication.

Experimental design

The author accepted my suggestion in its entirety and agreed with its revision.

Validity of the findings

The author accepted my suggestion in its entirety and agreed with its revision.

Additional comments

The author accepted my suggestion in its entirety and agreed with its revision.